# Bisindolylpyrrole Induces a Cpr3- and Porin1/2-Dependent Transition in Yeast Mitochondrial Permeability in a Low Conductance State via the AACs-Associated Pore

**DOI:** 10.3390/ijms22031212

**Published:** 2021-01-26

**Authors:** Masami Koushi, Rei Asakai

**Affiliations:** Department of Morphophysiology, Faculty of Pharmaceutical Sciences, Josai International University, 1 Gumyo, Togane, Chiba 283-8555, Japan; koushi@jiu.ac.jp

**Keywords:** *Saccharomyces cerevisiae*, mitochondrial permeability transition pore, bisindolylpyrrole, ADP/ATP carrier

## Abstract

Although the mitochondrial permeability transition pore (PTP) is presumably formed by either ATP synthase or the ATP/ADP carrier (AAC), little is known about their differential roles in PTP activation. We explored the role of AAC and ATP synthase in PTP formation in *Saccharomyces cerevisiae* using bisindolylpyrrole (BP), an activator of the mammalian PTP. The yeast mitochondrial membrane potential, as indicated by tetramethylrhodamine methyl ester signals, dissipated over 2–4 h after treatment of cells with 5 μM BP, which was sensitive to cyclosporin A (CsA) and Cpr3 deficiency and blocked by porin1/2 deficiency. The BP-induced depolarization was inhibited by a specific AAC inhibitor, bongkrekate, and consistently blocked in a yeast strain lacking all three AACs, while it was not affected in the strain with defective ATP synthase dimerization, suggesting the involvement of an AAC-associated pore. Upon BP treatment, isolated yeast mitochondria underwent CsA- and bongkrekate-sensitive depolarization without affecting the mitochondrial calcein signals, indicating the induction of a low conductance channel. These data suggest that, upon BP treatment, yeast can form a porin1/2- and Cpr3-regulated PTP, which is mediated by AACs but not by ATP synthase dimers. This implies that yeast may be an excellent tool for the screening of PTP modulators.

## 1. Introduction

The mitochondrial permeability transition pore (PTP) is a Ca^2+^-dependent non-specific hydrophilic channel with an exclusion size of 1.5 kDa within the inner mitochondrial membrane, which leads to mitochondrial membrane depolarization and swelling [1,2,3] and is associated with cell death under a variety of pathologies [4]. The key characteristics of this pore are conductance of approximately 1.2–1.5 nS and sensitivity to cyclosporin A (CsA), an inhibitor of cyclophilin D (CypD) [4,5,6,7,8]. The PTP has long been hypothesized to be formed by the voltage-dependent anion channel (VDAC) or the adenine nucleotide translocase (ANT); however, this hypothesis has been questioned based on gene knockout experiments with VDAC1/2/3 genes [9] and ANT1/2 genes [10]. Several reports have proposed that the mitochondrial ATP synthase directly involves pore formation through the membrane-embedded ring of its c subunits [11,12,13] or a putative channel structure in between two monomers of this enzyme [14,15]; however, these proposals have been challenged based on knockout studies with ATP synthase subunits in HAP1-A12 cells [16,17,18] and are still under debate [19,20]. The ANT can form bongkrekate (a specific ANT inhibitor)-sensitive, Ca^2+^-activated channels with conductance up to 0.6 nS with isolated reconstituted ANT [21,22]. Most recently, this ANT pore hypothesis was reevaluated because the mitochondria lacking the ATP synthase c-subunit formed a CsA-sensitive low conductance channel with bongkrekate sensitivity [23] and liver mitochondria from ANT1/ANT2/ANT4 triple knockout mice became highly resistant to Ca^2+^-induced PTP formation [24]. Notably, a report estimated the formation of only a few PTPs per mitochondrion during mitochondrial swelling [25]. The PTP also opens reversibly [26,27,28,29,30,31] but quite rarely in non-stress conditions [32], and transient pore opening may function as a Ca^2+^-releasing channel [26,29,30,31] and be of pathophysiological relevance to mitochondrial Ca^2+^ handling [33,34,35]. Like mammalian mitochondria, yeast mitochondria in *Saccharomyces cerevisiae* show swelling upon ethanol respiration [36] and Ca^2+^ transfer by calcium ionophore II (ETH129) [37]. Although it has been recognized that yeast PTP is not sensitive to CsA [36,37] or the CypD homolog Cpr3 [38], which can function as a chaperone in protein folding in the mitochondrial matrix through catalyzing peptidyl-prolyl cis-trans isomerization [39], we showed that the swelling of agar-embedded yeast mitochondria is sensitive to CsA/Cpr3 and mediated by endogenous matrix Ca^2+^ [40] (the level was comparable to that of rat liver mitochondria [41]). CsA-like PTP inhibitors were expected to be potential therapeutics for PTP-based diseases [42,43,44]; however, the prototype PTP inhibitor CsA failed in clinical trials [45,46]. Bisindolylmaleimide [47] and its derivative bisindolylpyrrole (BP) [48] are compounds that are cytoprotective against oxidative insult. In the preliminary experiments to elucidate the cytoprotective mechanism of these compounds, we found that the cytoprotective effect against Ca^2+^-mediated oxidative cell death in mammalian cells could be counteracted, paradoxically, by CsA or CypD ablation; therefore, we hypothesized that the mechanism was associated with the PTP. Indeed, we demonstrated that under specific conditions, BP acts on the PTP to trigger transient openings, leading to apoptosis, in a CypD- and VDAC1/2-dependent manner via the ANT-associated pore [49].

In the present study, we used yeast mutants to investigate whether ANT and ATP synthase play differential roles in PTP formation. To do this, we used BP as a probe to directly measure its effect on mitochondria in yeast mutants lacking all AACs (ANT homologs) or with defective ATP synthase dimerization, in addition to mutants deficient in porin1/2 or Cpr3. We found that BP can trigger CsA/Cpr3- and porin1/2-sensitive mitochondrial membrane depolarization. Interestingly, BP-induced depolarization was sensitive to bongkrekate and was blocked in a yeast mutant lacking all three AACs, while it was not affected in a strain with defective ATP synthase dimerization. In isolated mitochondria, BP induced depolarization but not calcein release, suggesting a low conductance PTP formation. Considering our previous data obtained with yeast mitochondria, which showed that the Ca^2+^-induced low conductance PTP channel was associated with ATP synthase dimers [40], we propose the existence of two types of the low conductance pores associated with either AACs or ATP synthase. 

## 2. Results

### 2.1. BP Induces Mitochondrial Depolarization in Yeast Cells, which Is Dependent on Cpr3

We explored whether BP induces pore opening in yeast cells. It is impossible to directly monitor PTP opening by loading calcein in in situ mitochondria; as previously reported, treatment with calcein-AM caused the preferential accumulation of calcein in the vacuoles [40]. Thus, we measured mitochondrial membrane potential by means of tetramethylrhodamine methyl ester (TMRM). As shown in Figure 1, we found that 5 μM BP could trigger mitochondrial depolarization progressively over a 4 h period. This depolarization was almost completely blocked by CsA as well as by Cpr3 deficiency [50]. These findings indicate that BP can induce CsA/Cpr3-dependent mitochondrial depolarization.

### 2.2. BP-Induced Mitochondrial Depolarization Depends on Porin 1 and Porin 2

Utilizing yeast mutant strains lacking either or both porin 1 or porin 2 (yeast homologs of VDAC) [51], we investigated whether these channels are associated with BP-induced depolarization. Δ*por1/2* cells doubly lacking porin 1 and porin 2 were refractory to BP; Δ*por1* and Δ*por2* were also resistant, although less effective relative to Δ*por1/2* (Figure 2). These suggest that BP-induced PTP formation requires both porin 1 and porin 2, in line with the involvement of VDAC1/2 in the BP-triggered transient PTP opening associated with apoptosis [49]. 

### 2.3. Pore Opening Is Mediated by AACs but Not ATP Synthase Dimers

Both ANT and ATP synthase are top candidates for CsA-sensitive PTP formation in mammalian mitochondria. To determine whether yeast homologs of these two molecules are also involved in BP-induced yeast PTP formation, we employed Δ*AAC1/2/3*, a mutant strain lacking all three AAC isoforms [52], and Δ*ATP20*, a strain essentially devoid of ATP synthase dimers due to the lack of a *g* subunit [38]. Δ*AAC1/2/3* cells showed no depolarization in response to BP, in contrast with the depolarization in parental cells (W303-1B) (Figure 3i), suggesting involvement of AACs in BP-induced permeability. To confirm this, we used the ANT-specific inhibitor bongkrekate [53]. The addition of 10 μM bongkrekate resulted in significant inhibition of depolarization at 4 h (Figure 3ii). The partial block by bongkrekate is consistent with previous reports of its effect on the PTP channel activity in patch-clamp experiments with mitoplasts [22,23,25]. 

Next, we examined the role of ATP synthase dimers. The mitochondria from Δ*ATP20* (Figure 4) and parental yeast cells (BY4743) (Figure 2) were equally susceptible to BP treatment, suggesting that ATP synthase dimers may not play an essential role in pore formation. The AACs’ possible involvement was tested in Δ*ATP20* cells using bongkrekate. This ANT inhibitor could significantly inhibit pore opening. These data support that AACs, rather than dimeric ATP synthase complex, may contribute to BP-induced permeabilization.

### 2.4. BP Releases TMRM but Not Calcein in Isolated Yeast Mitochondria

Using isolated yeast mitochondria that were preloaded with calcein, as previously described [40], we investigated the direct effect of BP on mitochondria. The membrane potential dissipated 5–10 min after treatment with BP, and this dissipation was sensitive to CsA and bongkrekate. In contrast, BP had no significant effect on the mitochondrial calcein signals over 10 min (Figure 5). These data suggest that BP-induced, CsA-dependent depolarization may be an AAC-mediated process in a low conductance state. 

## 3. Discussion

We demonstrated that treatment of yeast cells with BP can induce CsA/Cpr3-sensitive mitochondrial depolarization (Figure 1), and that this depolarization appears to be due to a low conductance permeability, as judged by the release of TMRM, but not of calcein, in isolated mitochondria (Figure 5). The depolarization effect of BP took a few hours in yeast cells, which is much slower than the depolarization observed in BP-treated HeLa cells [49]. This is likely due to the presence of the yeast cell wall. Indeed, isolated yeast mitochondria were shown to respond rapidly to BP (within 10 min). As demonstrated in cells (Figure 3) and isolated mitochondria (Figure 5), BP-induced pore opening is inhibited by bongkrekate, which does not affect the channel activity of ATP synthase [15], and is completely blocked in Δ*AAC1/2/3* cells, indicating the formation of an AAC-associated pore in response to BP. This yeast pore might be comparable to bongkrekate-sensitive low conductance substates in ATP synthase c-subunit-knockout mammalian mitochondria [23]. It should be noted that it is unlikely that the pore formation by BP depends on the mitochondrial Ca^2+^ accumulation since yeast mitochondria lack a Ca^2+^ uniporter [36], and the BP-induced depolarization in isolated yeast mitochondria was able to be induced by the absence of Ca^2+^ in the mitochondrial buffer used (Figure 5). In contrast, as shown previously with the mitochondria isolated from Δ*AAC1/2/3* cells, the absence of all three AACs has no effect on the formation of the CsA-sensitive, Ca^2+^-mediated high conductance pore in response to NADH respiration [40]. These data suggest that the yeast PTP may be mediated by at least two distinct molecules, depending on the pore inducers: the AACs and an unknown species requiring Cpr3, presumably the ATP synthase. The role of mammalian ATP synthase as a channel component of the PTP is disputed [19,20] and is questioned since *ρ*^0^ cells, where the major dimerizing component ATP6 is deficient, still have a functional PTP [54]. However, Δ*ATP20* mitochondria with defective ATP synthase dimerization are significantly resistant to Ca^2+^/ETH129-induced transition in CsA-sensitive low conductance permeability [40], indicative of the formation of an ATP synthase-associated pore. In contrast, BP-treated Δ*ATP20* cells show normally bongkrekate-sensitive depolarization (Figure 4), indicating the formation of an AAC-associated pore. Overall, these data suggest that a yeast CsA-sensitive PTP may be formed by the transition of either AACs or the dimer of ATP synthase into low conductance pores, depending on the applied pore stimulants (BP or Ca^2+^).

Further, it should be stressed that the yeast CsA/Cpr3-sensitive pore activation by BP in a low conductance state is dependent on porin1/2 in addition to the AACs, suggesting the existence of a PTP complex consisting of porin, AAC and Cpr3, in which the AAC functions as the core channel component. The involvement of these three proteins in yeast PTP activation is apparently comparable to the involvement of VDAC, ANT and CypD in the BP-induced transient PTP opening leading to apoptosis in HeLa cells [49]. The VDAC-ANT-CypD complex was previously demonstrated to function as a Ca^2+^-dependent, CsA-sensitive large pore channel when these three proteins were incorporated into liposomes [55], although VDAC was not essential for the specific binding of CypD to the ANT [56]. From these data, it seems that the well-known classic PTP model should be reevaluated, although it once was denied by genetic ablation studies.

Overall, yeast mitochondria may possess at least two distinct Cpr3-regulated permeability pathways through AACs and ATP synthase depending on the pore inducers. This conclusion indicates that the combined knockout of the ANT and ATP synthase genes could lead to the identification of the protein responsible for PTP formation. Finally, the present study implies that the CsA-dependent depolarization of yeast mitochondrial membrane potential could be an excellent indicator for screening small compounds to discover potential mammalian PTP modulators.

## 4. Materials and Methods

### 4.1. Yeast Strains and Culture and Induction of PTP Opening

The Saccharomyces cerevisiae strains of Δpor1/2 with the parent M3 (MATa, lys2, his4, trp1, ade2, leu2 and ura3), and Δcyp3/cpr3 (YM1372 with the Δcyp3::HIS3 mutation) with the parent YM1372 (MATa, ura3-52, his3-Δ200, ade2-101, lys2-801, trp1-901, tyr1-501, can1, gal80-538 and LEU2::GAL1-lacZ) [50,51] were a gift from Dr. M. Forte. W303-1B (MATα, ade2, his3, leu2, trp1, ura3 and can1) and ΔAAC1/2/3 lacking all three AAC-encoding genes (W303-1B with the aac1::LEU2, aac2::HIS3 and aac3::URA3 mutation) were a gift from Dr. J. Kolarov [52]. BY4743 (4741/4742), ΔATP20 (MATα, his3Δ1, leu2Δ0, lys2Δ0, ura3Δ0), Δpor1 and Δpor2 were purchased from Thermo Fisher Scientific (Waltham, MA, USA). M3 and Δpor1/2 cells were cultured in medium (0.3 g of yeast extract, 0.1 g of potassium phosphate, 0.1 g of NH_4_Cl, 0.05 g of CaCl_2_, 0.05 g of NaCl, 0.06 g of MgSO_4_, 0.03 mL of 1% FeCl_3_ and 4.8 mL of 42.5% lactic acid in 100 mL, adjusted to pH 5.0 by KOH). Δpor1, Δpor2, YM1372, Δcyp3/cpr3, BY4743 and ΔATP20 cells [38] were cultured in YPG medium (composed of 1% yeast extract, 2% peptone and 3% glycerol) (Thermo Fisher Scientific). W303-1B and ΔAAC1/2/3 were cultured in YPD (1% yeast extract, 2% peptone and 2% glucose). With the exception of Δpor1/2, yeast cell colonies were inoculated in 50 mL of medium overnight in an orbital shaker (230 rpm, 30 °C), 1 mL of which was added into 50 mL of medium, followed by incubation for 12–24 h, until cell concentrations reached 0.2–0.4 (A_600_). A colony of Δpor1/2 was cultured overnight in 5 mL of medium, which was added to 50 mL of medium and cultured for 12–24 h. BP treatment was conducted with yeast suspensions (1 × 10^6^ cells/mL), which were preincubated for 2 h with 50 nM TMRM (Molecular Probes, Eugene, OR, USA) in 50% basic medium (DMEM [Nissui, Tokyo, Japan], 10 mM HEPES-NaOH and 4 mM _L_-glutamine, pH 7.4, which was diluted twice with water).

### 4.2. Isolation of Yeast Mitochondria and Mitochondrial Calcein Loading and Activation of Yeast PTP

Small overnight cultures of 25 mL from a single colony were inoculated into 1.2 L cultures, followed by incubation for 12–24 h until cell concentrations reached 0.2–0.4 (A_600_). Using this culturing method, 0.5–1 g (wet weight) of yeast was usually harvested. Spheroplasts were prepared as previously described [40]. Briefly, yeast cells at a concentration of 0.15 g/mL (wet weight) in 0.1 M Tris-SO_4_ (pH 9.4) and 30 mM DTT were incubated for 10 min at 30 °C. After washing with 1.2 M sorbitol, the cells were suspended at 0.1 g/mL in 1.2 M sorbitol buffer containing 20 mM potassium phosphate (pH 7.4) and treated for 15 min with 100 U/mL of zymolyase 20T (20,000 U/g; Nacalai Tesque, Kyoto, Japan) at 30 °C. After washing, spheroplasts were vortexed to be ruptured in a buffer comprised of 0.6 M mannitol, 0.1 mM EDTA-K, 10 mM HEPES-K, pH 6.9 and 0.05% BSA (Wako, Osaka, Japan). After centrifugation at 3000× *g* for 3 min, the supernatants were spun down at 10,000× *g* for 5 min at 4 °C, and the resulting mitochondria pellets were resuspended in stock buffer (0.6 M mannitol, 0.1 mM EGTA-K, 10 mM HEPES-K, pH 6.9 and 0.05% BSA). Then, mitochondria (3 mg protein/mL) were treated for 1 h at 25 °C with 5 μM calcein-AM (Molecular Probes) in stock buffer supplemented with 2 mM Pi. After rinsing, they were stored in stock buffer on ice (10 mg protein/mL). Before inducing pore opening with BP with or without CsA or bongkrekate (BioMol, Plymouth, PA, USA), mitochondrial suspensions were energized at 25 °C with 5 mM succinate in 0.1 mL of mannitol buffer (0.3 M mannitol, 10 mM HEPES-K, pH 6.9) containing 100 nM TMRM. 

### 4.3. The Acquisition and Quantification of TMRM Fluorescent Signals 

After the induction of the PTP, laser scanning confocal microscopic images of yeast cells were acquired every 30 min for 4 h. TMRM fluorescence signals in yeast cells were calculated by subtracting the background signals (which were obtained from yeasts treated for 4 h with 0.5 μM carbonyl cyanide 4-(trifluoromethoxy) phenylhydrazone (FCCP), 2 h after preincubation with 50 nM TMRM) from signals in regions of interest (yeast whole cell bodies; manually applied), using the Olympus FV10–ASW software program (Olympus, Tokyo, Japan). TMRM and calcein signals from isolated yeast mitochondria were acquired every 30 s for 10 min, as described in detail [40]. Approximately 90% of mitochondria, which were more than 0.66 μm in diameter and stained with both calcein and TMRM, were subjected to the analysis. Fluorescent signals from two hundred randomly selected mitochondria were calculated by subtracting the background signals from those in regions of interest (0.34 μm^2^) of 3 pixels in diameter. The relationship between TMRM fluorescence and the mitochondrial membrane potential is described by the Nernst equation, and to linearize changes in mitochondrial membrane potential, TMRM fluorescence is presented on a logarithmic scale and background signals are expressed as 1 arbitrary unit (a.u.); levels under 5 a.u. were considered to represent complete depolarization.

### 4.4. Laser Scanning Confocal Microscopy

Fluorescence signals were monitored using an Olympus FV-1000 laser scanning confocal microscope (Olympus, Tokyo, Japan) equipped with a x60 oil immersion objective for cover glasses. Calcein was excited by a 488 nm argon-krypton laser, and fluorescence was detected using a 505–530 nm band-pass emission filter. TMRM was excited at 543 nm using a helium-neon laser, and fluorescence was detected using a 560–620 nm band-pass filter.

### 4.5. Statistical Analysis

All results are representative of at least three different mitochondrial isolations and at least three cell death experiments with HeLa cells. Data were statistically analyzed using a one-way or two-way ANOVA, followed by a multiple comparison analysis with Welch’s *t*-test with Bonferroni’s correction for bar graphs and Welch’s *t*-test for the time course.

## Figures and Tables

**Figure 1 ijms-22-01212-f001:**
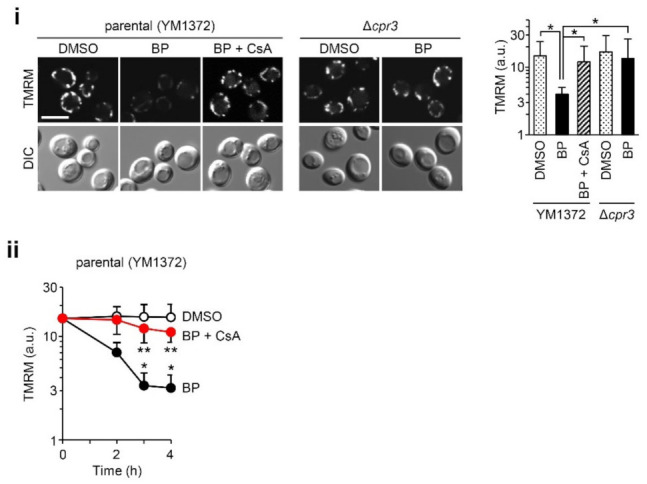
The effect of bisindolylpyrrole (BP) on the mitochondrial membrane potential in wild-type (YM1372) and Cpr3-deficient (Δ*cpr3*) yeast strains (*Saccharomyces cerevisiae*), pre-stained with 50 nM tetramethylrhodamine methyl ester (TMRM). (**i**) Left panels show confocal images for TMRM fluorescence signals at 4 h; representative fields were randomly imaged. Scale bar: 5 μm. Right panel shows the quantification of TMRM fluorescence signals (data represent the mean ± SD from 100–150 cells; * *p* < 0.001 by Welch’s *t*-test with Bonferroni correction). The potential levels under 5 a.u. obtained by treatment with 0.5 μM carbonyl cyanide 4-(trifluoromethoxy) phenylhydrazone (FCCP), an uncoupler, were considered to reflect depolarization. (**ii**) The time course of TMRM fluorescence over 4 h in the presence of DMSO (vehicle) or 5 μM BP with or without 3 μM cyclosporin A (CsA) in YM1372 (data represent the mean ± SD from 100–150 cells; * *p* < 0.001 relative to DMSO control; ** *p* < 0.001 relative to BP at the same time points, by Welch’s *t*-test).

**Figure 2 ijms-22-01212-f002:**
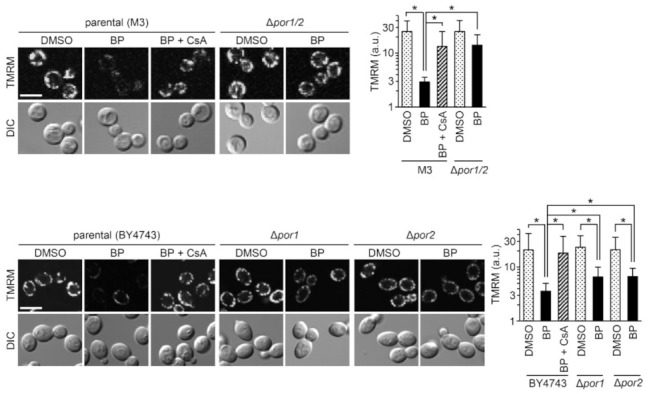
The effect of BP on the mitochondrial membrane potential in a mutant strain with double deficiency (Δ*por1*/*2*) and its parental strain (M3), and mutant strains lacking each of porin 1 (Δ*por1*) or porin 2 (Δ*por2*) and their parent strain (BY4743). Confocal images for TMRM fluorescence at 4 h (left panels) and their data, which represent the mean ± SD from 100–150 cells (* *p* < 0.001 in upper panel, * *p* < 0.005 in lower panel). Note the resistance of porin 1, porin 2, and porin/1/2 mutants to BP-induced depolarization. Scale Bars: 5 μm.

**Figure 3 ijms-22-01212-f003:**
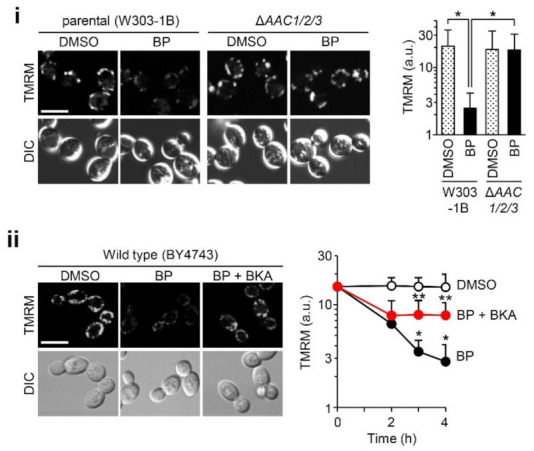
The involvement of AACs in BP-induced mitochondrial depolarization. (**i**) Yeast strains of the parental type (W303-1B) and deficiency of all three AACs (Δ*AAC1*/*2/3*) were treated with BP for 4 h. TMRM fluorescence images at 4 h (left panel) and quantification (right panel) (error bars represent the mean ± SD from 100–150 cells; * *p* < 0.001). (**ii**) BY4743 cells were treated by BP with or without 10 μM bongkrekate (BKA), a specific AAC inhibitor. Confocal images at 4 h (left panel) and their time courses over 4 h (right panel: data represent the mean ± SD from 100–150 cells; * *p* < 0.001 relative to DMSO control; ** *p* < 0.01 relative to BP at the same time points). The experimental conditions are shown in Figure 1. Scale Bars: 5 μm.

**Figure 4 ijms-22-01212-f004:**
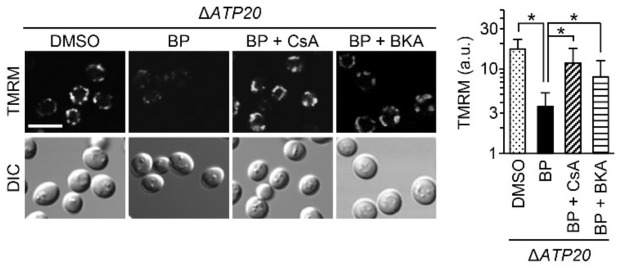
The effect of BP, with or without CsA or bongkrekate (BKA), on the mitochondrial membrane potential in Δ*ATP20*, a mutant strain with defective dimerization of ATP synthase. The effect of BP on its parental strain (BY4743) is shown in Figure 3. TMRM fluorescence images at 4 h (left panel) and quantification (right panel) (data represent the mean ± SD from 100–150 cells; * *p* < 0.001). Scale Bar: 5 μm.

**Figure 5 ijms-22-01212-f005:**
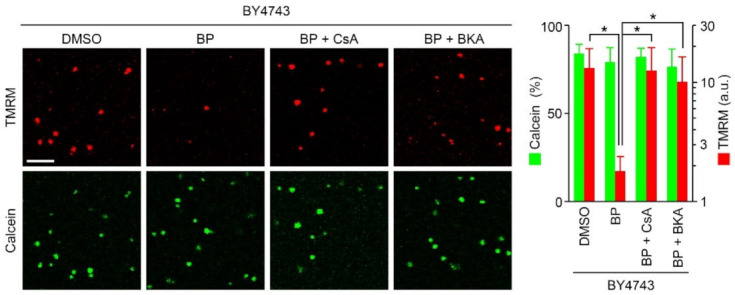
BP opens the CsA- and bongkrekate-sensitive low conductance channel in isolated yeast mitochondria, as indicated by the release of TMRM but not calcein. Isolated mitochondria from BY4743, preloaded with calcein (green) and labeled with TMRM (red), were energized at 25 °C with 5 mM succinate and treated with 5 μM BP in the presence or absence of 3 μM CsA or 10 μM bongkrekate. Representative confocal images at 10 min after BP treatment and quantification of TMRM signals (data represent the mean ± SD from 200 mitochondria; * *p* < 0.001). Scale bar: 5 μm.

## Data Availability

Data is contained within the article.

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
