# Peer review of "Bisindolylpyrrole Induces a Cpr3- and Porin1/2-Dependent Transition in Yeast Mitochondrial Permeability in a Low Conductance State via the AACs-Associated Pore"

_ijms, 2021, doi:10.3390/ijms22031212_

Round 1

Reviewer 1 Report

This paper by Koushi and Asakai investigates whether ANT (AAC) or ATP synthase participate in MPTP. The authors' approach is to induce MPTP via BP in wild-type mitochondria or mitochondria harboring ACC deletions or an ATP synthase subunit deletion. The authors further assess depolarization in isolated mitochondria. 

Overall, this paper is very well-written and easy to comprehend. The approaches are straight-forward, the data are quantified and appropriately analzyed, and the conclusions generally match the data presented. 

A strength of this paper is that it does not try to oversell the results. Instead, the approach was plainly stated: the authors were seeking to provide more evidence as to the molecular constituents of MPTP in yeast. The authors did, in fact, provide more evidence to the field in this context. This topic will be of general interest to the IJMS audience and the paper meets the stated standards of IJMS. 

Therefore, this reviewer recommends acceptance with no revisions.

However, there were two minor points of clarification that I encourage the authors to address prior to submitting their final draft to the editors:

  • Each figure must be referred to in the results. From the Results text, it is apparent that section 2.1 describes Figure 1 and section 2.2 describes Figure 2. However, there is no actual indication of this in the Results text. 
  • Related to Figure 4: The results text says "As shown in Figure 4, the mitochondria from ΔATP20 and parental yeast cells were equally susceptible to BP treatment..." However, no comparison to parental yeast is provided for this figure. 

Author Response

Thank you very much for the very positive and encouraging comments.

As suggested, we have included the phrases in the result section as follows: “As shown in Figure 1, we” (page 5, line 7); “Figure 2” (page 6, line 11).

The parental cell of ΔATP20 is BY4743. Thus, we revised the sentence (page 8, line 6 from the bottom) as follows: “The mitochondria from ΔATP20 (Figure 4) and parental yeast cells (BY4743) (Figure 2) were equally susceptible…...” 

Reviewer 2 Report

There is an ongoing dispute about the molecular nature of PTP. The present MS is an important contribution to the participation of adenine nucleotide carrier in PTP formation. 

The MS is compact and well written. Experiments are well designed, presentation is clear.

Comments and questions:

1) In the introduction authors write: "Bisindolylmaleimide [47] and its derivative bisindolylpyrrole (BP) [48] are compounds that are cytoprotective against oxidative insult; based on our investigations we suspected that the mechanism is linked to the PTP." Cytoprotection according to most of the interpretation does not mean that „BP acts on the PTP to trigger transient openings, leading to apoptosis”. I think these sentences create confusion, please clarify it.

2) I think authors should mention at least briefly the background experiments which proove that the phenomenon described here is indeed a PTP. The proofs are the Cyclosporin and bongkrekic acid sensitivity, however the timescale is relatively long, there is no calcein release.

3) Is there any proof for the calcium sensitivity of BP induced PTP?

4) A few words about the possible physiological role of Cpr3 would be useful.

Author Response

Thank you very much for pointing out these important points. We are happy to have the chance to submit our revised manuscript.

1) Our preliminary studies have shown that the cytoprotective effects were counteracted by CsA. We therefore revised the sentence (P. 4, line 6) as follows: "In the preliminary experiments to delineate the cytoprotective mechanism of these compounds, we found that the cytoprotective effect against Ca2+-mediated oxidative cell death in mammalian cells could be counteracted, paradoxically, by CsA or CypD ablation; we therefore hypothesized that the mechanism was associated with the PTP."

2)As pointed out, the timescale is relatively long. The depolarization effect of BP took a few hours in yeast cells, which is much slower than the depolarization observed in BP-treated HeLa cells [49]. This is likely due to the presence of the yeast cell wall. Indeed, isolated yeast mitochondria were shown to respond to BP rapidly (within 10 min). This discussion is now reflected in the manuscript (P. 10, line 7 from the bottom).

It is impossible to directly monitor PTP opening by loading calcein in in situ mitochondria, as treatment with calcein-AM caused the preferential accumulation of calcein in the yeast vacuoles [40]. This important point has now been added in the discussion section (P. 5, line 3).

3) It is unlikely that  the pore formation of BP depends on the mitochondrial Ca2+ accumulation, since yeast mitochondria lack a Ca2+ uniporter and the BP-induced depolarization in isolated mitochondria was able to be induced in the absence of Ca2+ in the mitochondria buffer used.  This comment is now included in the discussion (P. 11, line 2).

4)As suggested, we modified the sentence (P. 3, line 2 from the bottom) as follows: .....the CypD homolog Cpr3、which can function as a chaperone in protein folding in the mitochondrial matrix through catalyzing peptidyl-prolyl cis-trans isomerization. 

Round 2

Reviewer 2 Report

I have no further comments.